# An Empirical Study of the Influencing Factors of University–Enterprise Authentic Cooperation on Cross-Border E-Commerce Employment: The Case of Zhejiang

**Meiyu Fang** [1,*,†]  **and Zhe Zhu** [2,†]

1   School of International Business, Zhejiang International Studies University, Hangzhou 310023, China
2   School of Economics, Zhejiang University of Technology, Hangzhou 310023, China
*   Correspondence: myfang@zisu.edu.cn
†   These authors contributed equally to this work.

**Abstract:** This paper investigates the positive influencing factors and the degree of influence of authentic university–enterprise cooperation on the employability of cross-border e-commerce graduates. We used the literature analysis method and Delphi method to construct 12 factors affecting employability and put forward corresponding hypotheses. We tested the data and correlation analysis by SPSS software and used structural equation modeling for parameter estimation and path coefficient measurement to construct a relationship model of the factors influencing the employability of college students by cross-border e-commerce university–enterprise cooperation. The results show that the university–enterprise authentic cooperation has positive impacts on college students' employability in theoretical knowledge, learning, professional skills application, personal basic quality, and professional quality. With the rapid development of cross-border e-commerce, studying the factors that affect the employability of college students through university–enterprise cooperation can not only promote the further integration of industry and education but also further improve the employability of college students, thus promoting the development of social economy. Therefore, the research is of positive significance.

**Keywords:** university–enterprise authentic cooperation; employability; cross-border e-commerce; investigate; influencing factors; structural equation modeling

## 1. Introduction

Cross-border e-commerce has become a new driving force for the development of China's foreign trade. In recent years, the annual growth rate of cross-border e-commerce in China has remained above 20%. In most of the Chinese universities, cross-border e-commerce talent cultivation is actually a mode of university–enterprise authentic cooperation. The industry chain of cross-border e-commerce involves "customs, CIQ, foreign exchange, tax, industry and commerce, logistics, finance, legislation, service and platform", which is characterized by interdisciplinary integration and strong practicality. The cultivation of such talent can only be achieved through the all-round authentic cooperation between university and enterprises in the aspects of curriculum system, faculty team construction, and the cooperative education mechanism of government, industry, university, and research based on the integration of production and education, cooperation of government, industry, university and research. Only in this way can graduates meet the market demand to be cultivated.

Based on the analysis of the current situation of cross-border e-commerce development in Hangzhou and the employment demand of cross-border e-commerce enterprises, Hangzhou Cross-border E-commerce Comprehensive Experimental Zone vigorously funded undergraduate colleges and universities to carry out cross-border e-commerce talent training and funded more than 10 colleges and universities, such as Zhejiang International

Studies University, Zhejiang Gongshang University, and Zhejiang Financial College, with funds ranging from 1 million yuan to carry out cross-border e-commerce talent training; establish cross-border e-commerce business schools, cross-border e-commerce talent, ports and talent training bases; and establish a hierarchical sub-echelon system of management talents, entrepreneurial talents, and elite talents.

Data from the Ministry of Human Resources and Social Security of the People's Republic of China show that the number of college graduates in 2022 will reach 10.76 million, and the scale and increase in college graduates will exceed 10 million for the first time, which is a record high. In China's earliest institutions of higher learning that have implemented cross-border e-commerce talent cultivation, the established experimental classes have adopted the university–enterprise authentic cooperation, but other traditional classes have not adopted this model. According to the survey, more than 89.66% of the graduates in the class adopting the mode of university–enterprise authentic cooperation choose to work in cross-border e-commerce, while only 46.65% in the class without this teaching mode.

Hence, we began to consider what are the factors that influence the cross-border e-commerce talent cultivation through the university–enterprise authentic cooperation. Furthermore, what is their mechanism of influence? The purpose of this paper is to discuss the above issues, try to determine the influencing factors on employment, and make an empirical analysis of these influencing factors.

The following is a review of the conceptual framework of this article. First is the concept of university–enterprise cooperation. There are many definitions and explanations of university–enterprise cooperation from different perspectives by domestic and international scholars. Este and Patel complement the model of interaction between university researchers and industry, and they describe the interests of university–enterprise cooperation as concentrated in the field of cooperative marketing, including patents, licensing, spin-off companies, and derivative products, which mainly involve one-way knowledge flow [1].

Regarding the definition of university–enterprise cooperation, the World Association for Cooperative Education believes that university–enterprise cooperation is the combination of learning in the classroom and on the job, helping students to translate and apply their theoretical knowledge into practice, while bringing back to school problems and new ideas encountered in the real world of work, thus promoting teaching and learning in schools.

Some studies found through interviews that international university collaboration could contribute to international industry collaboration, and the positive internal mechanism and external policy environment of universities can play a positive role in regulating the motivation of individuals and enterprises in university–enterprise cooperation [2,3]. Wang et al. studied the game between universities and enterprises from a game theory perspective and included local governments in his examination, finding that joint games can achieve Pareto optimality [4]. Moreover, government support has a positive impact on the efficiency of knowledge transfer, with government financial investment playing a positive role in promoting university–enterprise collaboration [5]. The author points out that the sample of this study involves 30 provinces in China, and the research scope is relatively macro, so there is a lack of pertinence. Future research can narrow the scope of research to the urban level. Inspired by this, this article selects major universities in Hangzhou, Zhejiang Province, for research, focusing on the micro level.

University–enterprise cooperation is an increasingly important enterprise innovation model [6]. The cooperation between emerging industries and universities is more to promote the development of new knowledge. Enterprises expect to acquire the development of new knowledge and obtain scientific support for the development of new products, so they are more likely to cooperate with universities [7]. However, in emerging industry enterprises, continuous cooperation with universities after completion of projects is less common. This may have to do with the fact that students often act as mediators in these one-off collaborations: the relationship of trust between the university and the company that

collaborative projects require may not be possible on a single project basis [8]. Therefore, we began to think about what kind of university–enterprise cooperation can be regarded as authentic cooperation.

Here is the definition of authentic university–enterprise cooperation. Hargreaves defines "authentic" cooperation as a sustainable partnership based on the interaction between the two parties [9]. The partners are equal in such cooperation. He compared "authentic" cooperation with "factitious" cooperation and defined it as a form of implementing partnership generally managed by principals or business leaders.

Therefore, we define "university–enterprise authentic cooperation" as follows: university–enterprise authentic cooperation is a sustainable cooperation model established between university and enterprises, in which the principals of both parties participate in the management. All cooperation members cooperate in a balanced and mutually beneficial way, and in the process of cooperation, they generate new knowledge, develop new innovative models, cultivate practical and effective talents, and ultimately promote social development in a win–win situation.

Finally, there is the concept of employability. Harvey, Maher, and Graves point out that full employment ability can enable graduates to obtain more employment opportunities [10,11]. Yorke also believes that a certain degree of employability can make job seekers more likely to obtain employment opportunities and succeed in occupations. In 2006, he redefined the employability of graduates as a series of achievements in skills, understanding, and personal characteristics that make graduates more likely to obtain employment opportunities and succeed in the industry [12]. Through this series of definitions, it can be seen that employability not only refers to the result of seeking a job but also emphasizes the potential of individuals to develop and succeed in the industry field [13].

Scholars also divide graduates' employability into hard power and soft power [14,15]. Hard power mainly refers to some theoretical applications and skills based on professional knowledge. Soft power includes a series of personal attributes and accomplishments [16,17]. Keiper pointed out that employability requirements may vary from discipline to discipline, and no single employability framework is coherent, systematic, comprehensive, and specific [18]. Therefore, cross-border e-commerce employability can be defined as the knowledge and skills necessary to obtain employment opportunities and successfully contribute to cross-border e-commerce enterprises.

Many previous studies believed that internships can bring benefits to both students and enterprises. From the perspective of employability, internships can improve students' employability skills, thus improving employability and gaining favor from enterprises. Moreover, in the process of internship, students can obtain real employment experience, and enterprises can also find potential employees [19].

International orientation adds special value to graduates in the context of current international education, which is reflected in national skills and intercultural competence [20,21]. According to the above classification, we initially put forward four variables of cross-border e-commerce students' employability: variables in knowledge, variables in skills, variables in professional quality, and variables in quality.

Hills points out that learning how to learn, work, interact, and collaborate in a team is crucial to an individual's success in a competitive environment [22]. Some Finnish employers operating in China have also put forward some employment advantages besides professional skills, such as the ability to work independently, team spirit, sense of responsibility, adaptability, extroverted personality, initiative, etc. In the spirit of teamwork, there is also a consistent point of view in the research after the COVID-19 outbreak. Salas-Vallina, Ferrer-Franco, and Herrera believe that teamwork ability is an important soft power, and Dirani et al. also draw the view that the ability to cooperate with others is crucial in employability [23,24]. As mentioned above, under the background of international education, national skills and cross-cultural abilities add special value to graduates. Understanding and recognizing diversity is also to vital employability, which may become increasingly important in the next generation. Respect for diversity means understanding and respecting

different cultures and backgrounds [25]. In the cross-border e-commerce field, we classify it as the ability to identify with different cultures.

Ishengoma and Vaaland studied the possibility of university–enterprise cooperation to stimulate students' employability by investigating students, teachers, and employees from 20 oil and gas industries, which also provides ideas for our research [26].

Based on the review of the above-mentioned literature and combined with the characteristics of cross-border e-commerce majors, we put forward 12 indicators of the impact of cross-border e-commerce university–enterprise cooperation on college students' employment, which are as follows: basic professional knowledge, cross-professional knowledge, professional practice ability, business communication ability, foreign trade situation analysis ability, business development ability, cross-cultural communication ability, heterogeneous cultural identity ability, corporate culture creation ability, team cooperation ability, self-learning ability, and initiative.

## 2. Materials and Methods

### 2.1. Conceptual Model

This paper assumes that the relationship between the four potential variables is as follows: professional theoretical knowledge, application of professional skills, personal quality, and individual basic quality interact with each other and ultimately work together on students' employability. As shown in Figure 1.

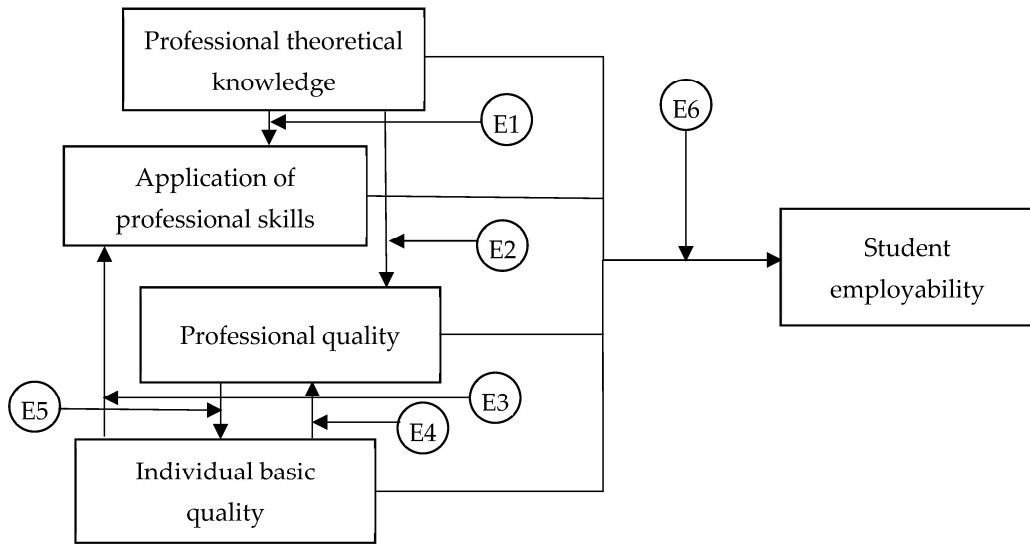

**Figure 1.** Conceptual model.

**E1.** *The p value of professional theoretical knowledge to application of professional skills;*

**E2.** *The p value of professional theoretical knowledge to professional quality;*

**E3.** *The p value of individual basic quality to application of professional skills;*

**E4.** *The p value of individual basic quality to professional quality;*

**E5.** *The p value of professional quality to individual basic quality;*

**E6.** *The p value of professional theoretical knowledge, application of professional skills, personal quality, and individual basic quality to employability.*

We verify the valid paths by the significance from E1 to E6. If the significance is greater than 0.05, the path is invalid and needs to be excluded; conversely, the path is valid.

*2.2. Delphi Method, Questionnaire Survey Analysis Method, and Their Applications in This Paper*

The Delphi method uses back-to-back communication to solicit the forecast opinions of the members of the expert group. After several rounds of consultation, the forecast opinions of the expert group tend to be concentrated, and, finally, the forecast conclusions in line with the future development trend of the market are made. To ensure the reasonableness of the indicators, we invited two teachers from cross-border e-commerce schools and one corporate mentor to conduct expert interviews, summarized their views on the impact of cross-border e-commerce university–enterprise authentic cooperation on students' employment, and adjusted the initial indicators based on the interview results.

This paper uses the method of questionnaire survey to collect the data needed for the research. With the help of questionnaire data analysis and empirical tests, the relationship between cross-border e-commerce university–enterprise authentic cooperation and the influencing factors of students' employability is examined. Using structured questionnaires to collect survey data, let them express their views on the importance of these factors in improving college students' employability. The ratings were performed on a Likert scale of 1–5, in which score 5 = strongly agree, score 4 = agree, score 3 = neutral, score 2 = disagree, and score 1 = strongly disagree.

*2.3. Delphi Method and Initial Index Selection*

After reading the literature and selecting the initial indicators, we obtained 12 indicators that cross-border e-commerce university–enterprise cooperation affects college students' employment. According to the four corresponding variables of cross-border e-commerce students' employability summarized above, we classify the 12 indicators into a table (see Table 1).

**Table 1.** Initial index and its classification.

| Ability | Initial Index |
|---|---|
| Variables in knowledge | A1: Basic Professional Knowledge |
| | A2: Cross-professional Knowledge |
| | A3: Professional Practice Ability |
| Variables in skills | B1: Business Communication Ability |
| | B2: Foreign Trade Situation Analysis Ability |
| | B3: Business Development Ability |
| Variables in professional quality | C1: Cross-cultural Communication Ability |
| | C2: Heterogeneous Cultural Identity Ability |
| | C3: Corporate Culture Creation Ability |
| Variables in quality | D1: Team Cooperation Ability |
| | D2: Self-learning Ability |
| | D3: Initiative |

This interview was conducted using semi-structured questions. The topic of the interview was to determine the impact of cross-border e-commerce cooperation between universities and enterprises on students' employment and related indicators. The four framework topics are the impact of cross-border e-commerce university–enterprise cooperation on students' knowledge, skills, professional qualities, and basic personal quality. The experts who participated in the interview analysed the initial indicators we selected and expressed their views on the classification of indicators. To avoid misunderstanding of the topics and questions in the interview, we explained the research purpose and relevant nouns before the interview. The main contents of the interview are as follows:

a. Describe the relationship between students' knowledge, skills, professional qualities, basic personal qualities, and students' employability.
b. Describe the rationality of the initial indicators corresponding to each variable, and propose modification suggestions.
c. Design the corresponding latent variables.

After the adjustment of expert semi-structured interviews, we have completed the determination of latent variables and initial indicators, as shown in Table 2.

**Table 2.** Latent variables and initial indicators.

| Ability | Latent Variable | Initial Indicator |
|---|---|---|
| Variables in knowledge | F1: Professional theoretical knowledge | A1: Basic Professional Knowledge<br>A2: Cross-professional Knowledge<br>A3: Professional Practice Ability |
| Variables in skills | F2: Application of professional skills | B1: Business Communication Ability<br>B2: Foreign Trade Situation Analysis Ability<br>B3: Business Development Ability |
| Variables in professional quality | F3: Professional quality | C1: Cross-cultural Communication Ability<br>C2: Heterogeneous Cultural Identity Ability<br>C3: Corporate Culture Creation Ability |
| Variables in quality | F4: Individual basic quality | D1: Team Cooperation Ability<br>D2: Self-learning Ability<br>D3: Initiative |

Based on the literature review and the determination of the indicators that affect the employability of college students, we put forward 12 assumptions about the factors that affect the employment of college students through university–enterprise cooperation:

(1) The Hypothesis of Influencing Factors of Cross-border E-commerce University–Enterprise Authentic Cooperation on Students' Professional Theoretical Knowledge:

**Ha1.** *Cross-border e-commerce university–enterprise authentic cooperation has a positive impact on students' learning of basic professional knowledge.*

**Ha2.** *Cross-border e-commerce university–enterprise authentic cooperation has a positive impact on students' expansion of cross-professional knowledge.*

**Ha3.** *Cross-border e-commerce university–enterprise authentic cooperation has a positive impact on students' in-depth mastery of professional theory.*

(2) The Hypothesis of Influencing Factors of Cross-border E-commerce University–Enterprise Authentic Cooperation on Students' Application of Professional Skills:

**Hb1.** *Cross-border e-commerce university–enterprise authentic cooperation has a positive impact on improving students' business communication ability.*

**Hb2.** *Cross-border e-commerce university–enterprise authentic cooperation has a positive impact on improving students' foreign trade situation analysis ability.*

**Hb3.** *Cross-border e-commerce university–enterprise authentic cooperation has a positive impact on improving students' business development ability.*

(3) The Hypothesis of Influencing Factors of Cross-border E-commerce University–Enterprise Authentic Cooperation on Students' Professional Quality:

**Hc1.** *Cross-border e-commerce university–enterprise authentic cooperation has a positive impact on cultivating students' cross-cultural communication ability.*

**Hc2.** *Cross-border e-commerce university–enterprise authentic cooperation has a positive impact on cultivating students' heterogeneous cultural identity ability.*

**Hc3.** *Cross-border e-commerce university–enterprise authentic cooperation has a positive impact on cultivating students' corporate culture creation ability.*

(4) The Hypothesis of Influencing Factors of Cross-border E-commerce University–Enterprise Authentic Cooperation on Students' Individual basic qualities:

**Hd1.** *Cross-border e-commerce university–enterprise authentic cooperation has a positive impact on cultivating students' team cooperation ability.*

**Hd2.** *Cross-border e-commerce university–enterprise authentic cooperation has a positive impact on cultivating students' self-learning ability.*

**Hd3.** *Cross-border e-commerce university–enterprise authentic cooperation has a positive impact on cultivating students' initiatives.*

*2.4. Questionnaire Survey*

The research objects of this paper are students majoring in cross-border e-commerce in universities in Hangzhou and employees of enterprises in the cross-border e-commerce industry. Relevant indicators and item designs are shown in Table 3.

**Table 3.** Observed variables and item designs.

| Ability | Latent Variable | Observed Variable | Questionnaire Items [1] |
|---|---|---|---|
| Knowledge | F1: Professional theoretical knowledge | A1: Basic Professional Knowledge | a1 |
| | | A2: Cross-professional Knowledge | a2 |
| | | A3: Professional Practice Ability | a3 |
| Skills | F2: Application of professional skills | B1: Business Communication Ability | b1 |
| | | B2: Foreign Trade Situation Analysis Ability | b2 |
| | | B3: Business Development Ability | b3 |
| Professional quality | F3: Professional quality | C1: Cross-cultural Communication Ability | c1 |
| | | C2: Heterogeneous Cultural Identity Ability | c2 |
| | | C3: Corporate Culture Creation Ability | c3 |
| Quality | F4: Individual basic quality | D1: Team Cooperation Ability | d1 |
| | | D2: Self-learning Ability | d2 |
| | | D3: Initiative | d3 |

[1] Notes: a1: Cross-border e-commerce university–enterprise cooperation can help students learn and understand basic professional knowledge. a2: It can help students contract and learn more cross-disciplinary knowledge. a3: It can help students master professional theory and knowledge in depth. b1: It can improve students' business communication ability. b2: It can improve students' foreign trade situation analysis ability. b3: It can improve students' business development ability. c1: It helps improve students' intercultural communication abilities. c2: It can improve students' heterogeneous cultural identity ability. c3: It can positively affect students' ability to learn corporate culture. d1: It can improve students' ability to analyze and solve problems. d2: It can improve students' team cooperation ability. d3: It has a positive impact on students' self-learning ability.

Since the purpose of this survey is to obtain the data needed to study the influencing factors of college students' employability and lay the foundation for empirical analysis, we designed the research questionnaires for students and enterprises respectively to conduct the survey. Hangzhou is the first city in China to carry out a comprehensive experimental zone for cross-border e-commerce, and Zhejiang International Studies University is a model institution for cross-border e-commerce talent training in China, which is one of the earliest universities in China to develop cross-border e-commerce majors, and the companies docked with them in the double-selection meeting also include a large number of cross-border e-commerce enterprises in Zhejiang. Hence, the students and enterprises are representative of the cross-border e-commerce profession and industry in Zhejiang, and since both are from the double-selection, the two types of data are also comparable. Taking into account the representativeness of the sample and the convenience of the survey, a random sample survey was conducted among the graduating class of students attending the Zhejiang International Studies University Double Elective Meeting, including students majoring in e-commerce (cross-border direction), computer science, international business, and financial engineering and students from the "3 + 1" experimental class. A total of 209 questionnaires were distributed, and 208 valid questionnaires were recovered, with a recovery rate of 99.52%.

In the enterprise survey, employees were selected from the enterprises participating in the Zhejiang International Studies University Double Election Conference to conduct

a random sampling survey. We have made a statement at the beginning of each survey questionnaire: we will keep the information strictly confidential and promise to use it only for scientific research and policy recommendations, not for other purposes. All respondents answered the questionnaire after accepting the commitment.

## 3. Results

### 3.1. Descriptive Statistical Analysis, Reliability and Validity Test, and Correlation Test

The overall average values of the three aspects of professional theoretical knowledge are 4.14, 4.15, and 4.12, respectively, which are all between 4 and 5 and between agreed and very agreed. The overall average score is 4.14, which shows that the students surveyed believe that cross-border e-commerce university–enterprise cooperation has a greater impact on the employability of college students in terms of knowledge. It is believed that cross-border e-commerce university–enterprise cooperation can help students connect and learn more cross-professional knowledge.

The overall averages of the three aspects of professional skills application are 4.14, 4.14, and 4.12, respectively, which are all between 4 and 5 and between agreed and very agreed. The overall average score is 4.13, which indicates that the school believes that cross-border e-commerce university–enterprise cooperation has a greater impact on college students' employability in terms of skills.

The overall averages of the three aspects of professional quality are 4.11, 4.09, and 4.11, respectively, which are all between 4 and 5 and between agreed and very agreed. The overall average score is 4.10, which indicates that students believe that cross-border e-commerce university–enterprise cooperation has a greater impact on college students' employability in terms of professional quality.

The overall averages of the three aspects of individual basic quality are 4.09, 4.10, and 4.13, respectively, which are all between 4 and 5, and between agreed and very agreed. The overall average score is 4.11, which shows that the students surveyed believe that cross-border e-commerce university–enterprise cooperation has a greater impact on college students' employability in terms of quality.

To make the results of the survey more convincing and to ensure the credibility and effectiveness of the next research, we need to conduct a reliability test and validity tests on the questionnaire. The test tool is SPSS25.0. The Cronbach's $\alpha$ coefficient of the enterprise and student survey data in this paper is greater than 0.9, indicating that the internal consistency of the scale is very high. Additionally, the KMO of students and enterprises in this paper are 0.939 and 0.951, respectively, with significant Sig. < 0.05 (i.e., $p < 0.05$), which is suitable for factor analysis.

### 3.2. Construction of Structural Equation Modeling and Estimation of Results Using Standardized Data

The structural equation model (SEM) is selected as the main tool for the empirical test in this paper.

In Table 4, the value of the modified model $X^2/df$ is 2.902, which meets the standard; the value of PGFI is 0.502, greater than 0.5, which meets the standard; as for GFI, GFI, NFI, CFI, IFI, and TLI, these indicators are greater than 0.9, in line with the standard. After removing the two paths, the structural equation model of the influencing factors of college students' employability becomes an unsaturated model. Therefore, some of the modified fitness indexes are slightly worse than those before modification, but they are all within the range of successful fitting.

In the measurement model of potential variable "professional theoretical knowledge", the factor loads of observable variables are 0.940, 0.948, and 0.965, respectively. It can be seen that the three observed variables all pass the significance test. Similarly, in the measurement of the potential variable "application of professional skills", the factor loads of the three observed variables are found to be 0.943, 0.934, and 0.979, respectively, all of which pass the significance test. In the measurement of the potential variable "personal

basic quality", the factor loads of the three observed variables are found to be 0.926, 0.963, and 0.927, respectively, which also pass the significance test. In the measurement of the potential variable "personal basic quality", the factor loads of the three observed variables are found to be 0.920, 0.957, and 0.978, respectively, which also pass the significance test. The above factor loads are shown in Table 5.

**Table 4.** Model fitness index.

| Indicator Name | Fitting Result | Reference Standard | Result |
|:---:|:---:|:---:|:---:|
| $X^2/df$ | 2.902 | <3 | pass |
| GFI | 0.910 | >0.9 | pass |
| PGFI | 0.502 | >0.5 | pass |
| NFI | 0.975 | >0.9 | pass |
| CFI | 0.983 | >0.9 | pass |
| IFI | 0.983 | >0.9 | pass |
| TLI | 0.974 | >0.9 | pass |

**Table 5.** Measures model factor load and significance.

| Measure Relation | | | Nonstandard Factor Load | Significant | Normalized Factor Load |
|:---:|:---:|:---:|:---:|:---:|:---:|
| a1 | <− | A1 | 1.000 | | 0.940 |
| a2 | <− | A2 | 0.990 | *** | 0.948 |
| a3 | <− | A3 | 1.006 | *** | 0.965 |
| b1 | <− | B1 | 1.000 | | 0.943 |
| b2 | <− | B2 | 0.980 | *** | 0.934 |
| b3 | <− | B3 | 1.032 | *** | 0.979 |
| c1 | <− | C1 | 1.000 | | 0.927 |
| c2 | <− | C2 | 1.052 | *** | 0.957 |
| c3 | <− | C3 | 1.066 | *** | 0.978 |
| d1 | <− | D1 | 1.000 | | 0.926 |
| d2 | <− | D2 | 1.051 | *** | 0.963 |
| d3 | <− | D3 | 1.002 | *** | 0.927 |

Note: Compiled from the AMOS report, the input data is the original data; Among them, "***" represents significant at the 0.001 level.

## 4. Discussion

### 4.1. Parameter Estimation and Path Coefficient Measure

Using the structural equation model and AMOS statistical analysis software, this paper empirically tests the relationship between the above-mentioned four aspects of ability and their elements and the employability of college students. To some extent, the path coefficient reflects the degree to which one variable affects another.

The results indicated that all 12 hypotheses proposed in this paper were tested. In addition, empirical tests based on structural equation modelling also found significant correlations among the competencies, such as a significant positive correlation between professional theoretical knowledge and professional literacy with a correlation coefficient of 0.962, which is a strong correlation; a significant positive correlation also exists between professional literacy and professional skills application with a correlation coefficient of 0.763, which is a strong correlation; the correlation coefficient between professional theoretical knowledge and professional skills application is 0.223, and it is significant at the level of 0.005, which is a weak correlation.

For the ability of knowledge, the path coefficient of variable A3 (master professional theory and knowledge in depth) to variable F1 (professional theoretical knowledge) is the largest; the standardized path coefficient is 0.965 and is significant at the level of 0.001; the path coefficient of variable A1 (professional basic knowledge) to variable F1 (professional theoretical knowledge) is the smallest, and the normalized path coefficient is 0.940, which is significant at the level of 0.001.

For quality variables, variable B2 (team spirit) has the largest path coefficient on variable F2 (individual basic competence) with a standardized path coefficient of 0.962 and is significant at the 0.001 level, while variable B1 (ability to analyse and solve problems) and variable B3 (initiative) have the same path coefficient on variable F2 (basic personal qualities) with standardized path coefficients of 0.926, and both were significant at the 0.001 level.

For the variables of professional quality, the path coefficient of variable C3 (corporate culture learning ability) to variable F3 (professional accomplishment) is the largest; the standardized path coefficient is 0.977, which has passed the significance test at the level of 0.001, while variable C1 (cross-cultural communication ability) has the smallest path coefficient to variable F3 (professional accomplishment); the standardized path coefficient is 0.922, which is significant at the level of 0.001.

For the ability of skills, the path coefficient of variable D2 (analysis of foreign trade situation) to variable F4 (application of professional skills) is the largest, and the standardized path coefficient is 0.979, which has passed the significance test at the level of 0.001; the path coefficient of variable D1 (professional basic knowledge) to variable F4 (professional skill application) is the smallest, and the normalized path coefficient is 0.934, which is significant at the level of 0.001. The results show that the 12 hypotheses proposed in this paper are all verified.

The results indicated that all 12 hypotheses proposed in this paper were tested. In addition, empirical tests based on structural equation modelling also found significant correlations among the competencies. For example, there is a significant positive correlation between professional theoretical knowledge and professional accomplishment, the correlation coefficient is 0.962, which is a strong correlation; there is also a significant positive correlation between professional accomplishment and application of professional skills; the correlation coefficient is 0.763, which is a strong correlation. The correlation coefficient between professional theoretical knowledge and application of professional skills is 0.223, which is significant at the level of 0.005 and is a weak correlation.

*4.2. Analyze the Impact Mechanism and Verify the Theoretical Assumptions Mentioned Above*

Through parameter estimation and path coefficient measurement, this paper constructs a relationship model of the influencing factors of cross-border e-commerce university–enterprise cooperation on college students' employability. Through the analysis of the influence mechanism, the above theoretical assumptions can be verified as follows.

First, in terms of knowledge, cross-border e-commerce university–enterprise cooperation has a positive impact on college students' professional knowledge, cross-professional knowledge, and in-depth mastery of professional theory. Therefore, cross-border e-commerce university–enterprise cooperation has a significant positive impact on students' professional theoretical knowledge, assuming that Ha1, Ha2, and Ha3 are verified. Secondly, in terms of skills, students and enterprises believe that their business development ability, business communication ability, and foreign trade situation analysis ability can be significantly improved in cross-border e-commerce university–enterprise cooperation, assuming that Hb1, Hb2, and Hb3 are also verified. In terms of professional quality, cross-border e-commerce university–enterprise cooperation has a positive impact on students' cross-cultural understanding ability, heterogeneous cultural identity ability, and corporate culture learning ability, assuming that Hc1, Hc2, and Hc3 are also verified. Finally, in terms of personal quality, the data show that cross-border e-commerce university–enterprise cooperation has a positive impact on students' ability to analyse and solve problems, team spirit, and initiative. Therefore, it is assumed that Hd1, Hd2, and Hd3 are also verified.

With the help of a structural equation model, this paper tests the relationship between cross-border e-commerce university–enterprise cooperation and the influencing factors of college students' employment. This paper clarifies the mechanism and path of cross-border e-commerce university–enterprise cooperation on various elements, finds out the significant correlation between each employment ability, and points out the direction of

how to build an authentic university–enterprise cooperation, how to improve college students' employment ability through an authentic university–enterprise cooperation, and how to adjust the current form of university–enterprise cooperation.

Although the above assumptions have been verified, there are still differences in relevance for different abilities and different aspects of the same ability. Students and enterprises have reached a consensus on some aspects but also expressed different views on some aspects. The existence of these differences from the side reflects that students and enterprises have different views on the current university–enterprise cooperation from different perspectives, which is reflected in their different roles, benefits, and contradictions in the university–enterprise cooperation.

Based on the above literature review, the complexity and challenges of university–enterprise cooperation are also reflected in the different concerns and expected results of stakeholders. In addition, students' answers also reflect the university's concerns in some aspects.

Therefore, on the basis of the results of this empirical analysis model, we need to discuss in depth which aspects universities and enterprises have reached a consensus on and which aspects have different views on the issue of university–enterprise cooperation to put forward relevant suggestions to promote the further development of the current university–enterprise cooperation, promote the further deepening of the integration of production and education, and realize the real university–enterprise cooperation.

*4.3. The Model of University–Enterprise Authentic Cooperation Affecting Employment*

In the model modification stage, because the degree of freedom of the initial model is greater than 3, the correction line is added to the model according to the relevant data in the Modification Indices in the output results. The following explains the added correction line from the meaning of residual.

The corresponding topics of E1 and E6 are "Cross-border e-commerce university–enterprise cooperation can help students learn and understand basic professional knowledge" and "Cross-border e-commerce university–enterprise cooperation can positively affect students' ability to learn corporate culture ". The standardized path coefficient is $-0.286$, which shows negative correlation. Good mastery of professional basic knowledge may reduce the flexible learning ability of corporate culture due to too much emphasis on basic knowledge. Moreover, some enterprises may focus on cultivating students' ability to combine theory and practice but neglect to guide students to learn the spirit of corporate culture.

The corresponding topics of E2 and E10 are "Cross-border e-commerce university–enterprise cooperation can help students contract and learn more cross-disciplinary knowledge " and "Cross-border e-commerce university–enterprise cooperation can help improve students' business communication ability ". The standardized path coefficient is $-0.408$, which shows negative correlation. This may be because the scope of interdisciplinary knowledge is wide, and business development ability is only part of it. In the process of helping students contact and learn more interdisciplinary knowledge, students tend to focus on some new fields, which will reduce their attention to the improvement of business development ability in this field.

The corresponding topics of E7 and E9 are "Cross-border e-commerce university–enterprise cooperation has a positive impact on students' self-learning ability " and "Cross-border e-commerce university–enterprise cooperation can improve students' ability to analyze and solve problems". The standardized path coefficient is 0.369, which shows a positive correlation. This is easy to understand, and stronger self-learning ability and problem-solving ability can play a role of mutual promotion; E10 and E11 correspond to business development and business communication, respectively, and the standardized path coefficient is 0.378, which shows a positive correlation. This is because business development and business communication capabilities are also complementary and interactive. Among them, in the residual item of initiative (E7), the part that cannot be explained by

personal basic quality can be explained by professional quality (E14), because initiative can also be a part of professional quality.

By fitting the hypothetical model and revising the model, the influencing factor model of college students' employability under the authentic cooperation between university and enterprise is obtained, as shown in Figure 2.

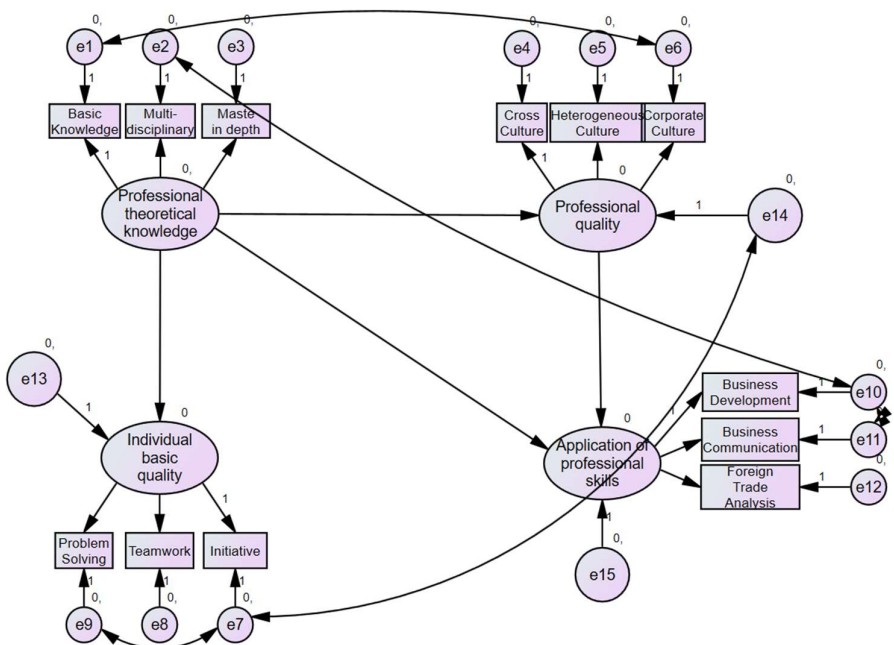

**Figure 2.** The influencing factor model.

The relationship between the four potential variables is as follows: professional theoretical knowledge and application of professional skills, personal quality, and professional accomplishment have mutual influence. To make the model diagram more concise and beautiful, some of the names of the observed variables are abbreviated—for example, "basic knowledge" should be "university–enterprise cooperation can help students learn and understand professional basic knowledge", "cross-cultural communication" should be "university–enterprise cooperation can help improve students' cross-cultural communication ability", "problem solving" should be "university–enterprise cooperation can help improve students' ability to analyze and solve problems", etc. The explanation of the correction line is shown in the model correction section above.

## 5. Conclusions

With the rapid development of cross-border e-commerce, today's market increasingly needs application-oriented talents with strong practical ability and high comprehensive quality. Under this trend, cross-border e-commerce university–enterprise cooperation becomes more and more important. In the talent training system of universities, theoretical teaching and practical teaching should be complementary and inseparable. However, the talent cultivation mode of most ordinary colleges and universities is basically based on theoretical mode cultivation, and the practical aspect cultivation is very little. In this paper, through the research and analysis of China's cross-border e-commerce talent cultivation, we obtain the influencing factors of university–enterprise authentic cooperation on college students' employment, which can provide more targeted reference for better university–enterprise cooperation so as to promote the further integration of industry-education.

### 5.1. Results

The research has drawn the following results and conclusions:

(1) Cross-border e-commerce university–enterprise cooperation can indeed improve college students' employability in terms of knowledge, technology, professional quality, and personal quality.

(2) Professional theoretical knowledge, application of professional skills, professional quality, and basic personal quality are the main competency modules that constitute the employability of college students. Authentic cooperation between cross-border e-commerce enterprises and enterprises has a positive and significant impact on these competencies. This shows that the greater the impact of cross-border e-commerce university–enterprise cooperation on these factors is, the stronger the employability of college students is.

(3) From the perspective of impact path, professional theoretical knowledge has a direct impact on the application of professional skills, professional accomplishment, and basic personal qualities, while professional accomplishment has a direct impact on the application of professional skills; judging from the degree of influence, professional theoretical knowledge has the most profound influence on individual basic quality, followed by professional quality, and the least influence on the application of professional skills.

(4) It is necessary to strengthen the cultivation of cross-professional knowledge and provide strong support for strengthening the cultivation of cross-professional knowledge in university–enterprise cooperation by jointly developing e-commerce cross-professional characteristic courses with enterprises, exploring teaching methods based on cross-professional cooperation, and building a shared professional teaching resource management platform.

The research in this paper can not only provide a valuable references for improving college students' employability but also provide a reference for promoting the development of university–enterprise cooperation.

### 5.2. Limitations of the Study

This study constructs a more systematic and complete model of the factors influencing college students' employability and empirically analyzes the factors influencing the employment of college students by university–enterprise authentic cooperation so as to provide a more targeted reference for better university–enterprise cooperation and promote the further integration of industry and education.

However, there are still some shortcomings in the study. Although in the construction of the research scale we refer to many domestic and foreign scholars' relevant literature on measurement items and employment ability indicators, the selection of specific indicators can be further detailed. Additionally, the research sample has certain limitations. In this empirical study, although students from different majors and personnel from different enterprises are covered as much as possible, samples from different regions are relatively scarce. If the number of samples from regions and universities is further improved, the validity of the study may be further enhanced. Therefore, in the future research, we should continue to expand the sample range in areas such as regions and student types to make the research more rigorous. It is also possible to conduct empirical studies for different majors and different levels of institutions and to subdivide the results so as to provide a very detailed and more specific reference for enriching the theoretical results of college students' employability.

**Author Contributions:** Conceptualization, M.F. and Z.Z.; methodology, M.F. and Z.Z.; software, M.F. and Z.Z.; validation, Z.Z.; formal analysis, M.F. and Z.Z.; investigation, M.F.; resources, M.F. and Z.Z.; data curation, M.F. and Z.Z.; writing—original draft preparation, M.F. and Z.Z.; writing—review and editing, M.F. and Z.Z.; visualization, Z.Z.; supervision, M.F.; project administration, M.F.; funding acquisition, M.F. All authors have read and agreed to the published version of the manuscript.

**Funding:** This research was funded by the Ministry of Education, Humanities, and Social Sciences Research Project Fund of China (No.14YJAZH018), Zhejiang Basic Public Welfare Research Project (No: LGF19G020002), the key projects of Pujiang cross-border e-commerce in 2023.

**Data Availability Statement:** The dataset can be accessed upon request.

**Conflicts of Interest:** The authors declare no conflict of interest.

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
