# Peer review of "An Empirical Study of the Influencing Factors of University–Enterprise Authentic Cooperation on Cross-Border E-Commerce Employment: The Case of Zhejiang"

_sustainability, doi:10.3390/su15086993_

Round 1

Reviewer 1 Report

Thanks for submitting this interesting study. Certainly, this topic is worth further research. Nevertheless, a few things worried me whilst reviewing your papers.

1-      The gap should be based on previous papers. It would be useful to make this more obvious. Clearly, state which paper made calls for such a study.

2-      Do you have ethical approval for this research? If yes, state it in this methodology section.

3-      I was wondering about your sample size, 208! So, does your sample represent all the students or enterprises worldwide? If it is all the students worldwide, how can you ensure that your sample size represents all students worldwide!! Or is it specific to a certain geographical area or country? Please state which geographical or country in which this study was conducted!! As students’ characteristics might differ from country to country, and state how you have achieved validity and reliability, you might also need to reflect that in the title.

4-      What are the practical and theoretical implications for your paper? Please state these before and after the conclusion section. 

Author Response

Response to Reviewer 1 Comments

Dear Reviewer:

We are very grateful to your comments for the manuscript. According with your advice, we amended the relevant part in manuscript. Some of your questions were answered below.

Point 1: The gap should be based on previous papers. It would be useful to make this more obvious. Clearly, state which paper made calls for such a study.

Response 1: We are grateful for the suggestion. Our research is inspired by the current situation of cross-border e-commerce university-enterprise cooperation, and an article in the journal of Education and Training titled Can university-industry linkages stimulate student employability also provides ideas for our research. The content in red font on lines 151-153 is a more detailed supplement.

Point 2: Do you have ethical approval for this research? If yes, state it in this methodology section.

Response 2: Thank you very much for your suggestion, which is very important for our research. In fact, we have made a statement at the beginning of each survey questionnaire: we will keep the information strictly confidential and promise to use it only for scientific research and policy recommendations, not for other purposes. All respondents answered the questionnaire after accepting the commitment. We have made supplementary explanations in lines 282-285 of the text, marked in red.

Point 3: I was wondering about your sample size, 208! So, does your sample represent all the students or enterprises worldwide? If it is all the students worldwide, how can you ensure that your sample size represents all students worldwide!! Or is it specific to a certain geographical area or country? Please state which geographical or country in which this study was conducted!! As students’ characteristics might differ from country to country, and state how you have achieved validity and reliability, you might also need to reflect that in the title.

Response 3: Thanks for your comments, the discussion regarding this question is presented following: The subjects of this survey are undergraduates majoring in cross-border e-commerce in universities in Hangzhou and enterprises in the cross-border e-commerce industry, which is mentioned on lines 273-278.

Since the targets of this survey are college students of cross-border e-commerce majors and enterprises of cross-border e-commerce industry in Hangzhou undergraduate colleges and universities, we designed the research questionnaires for students and enterprises respectively to conduct the survey. Hangzhou is the first city in China to carry out a comprehensive experimental zone for cross-border e-commerce, and Zhejiang International Studies University is a model institution for cross-border e-commerce talent training in China. The research subjects we selected were students and enterprises participating in the double-selection meeting of Zhejiang International Studies University, which is one of the earliest universities in China to develop cross-border e-commerce majors, and the companies docked with them in the double-selection meeting also include a large number of cross-border e-commerce enterprises in Zhejiang. Therefore, the "representativeness" mentioned here means that the students and enterprises are representative of the cross-border e-commerce profession and industry in Zhejiang, and since both are from the double-selection, the two types of data are also comparable. We clarify "representativeness" more clearly in lines 262-273 with red font.

Point 4: What are the practical and theoretical implications for your paper? Please state these before and after the conclusion section.

Response 4: We have supplemented the text according to the comments. The practical and theoretical implications of this article are adjusted and supplemented before and after the conclusion section, in lines 503-514 and lines 540-542, marked in red.

Reviewer 2 Report

1. Conceptual framework must be improved.

2. Conclusion part compared can be improved.

Author Response

Response to Reviewer 2 Comments

Dear Reviewer:

Thank you for your comments, and our reply is as follows:

Point 1: Conceptual framework must be improved.

Response 1: Thank you very much for your suggestion, which is very important for our research. We have strengthened the conceptual framework based on the original one and further clarified the concepts of university-enterprise cooperation, university-enterprise authentic cooperation and employability. We added the concept of university-enterprise cooperation in lines 66-77, and added articulation words in lines 98 and 110, thus making our conceptual framework more logical.

Point 2: Conclusion part compared can be improved.

Response 2: We have supplemented the text according to the comments. The practical and theoretical implications of this article are adjusted and supplemented before and after the conclusion section, in lines 503-514 and lines 540-542, marked in red.

Reviewer 3 Report

I appreciate the efforts

However please address the following concerns

i) Rewrite the abstract. It should clearly state why have you taken up the work, the research design description and the results implications

ii) Development of the constructs and hypothesis needs to the logical and adequately supported by arguements and LR

iii) Discussions on results needs to be strengthened

iv) limitations and implications need to be elaborated

Author Response

Response to Reviewer 3 Comments

Dear Reviewer:

Thank you for your precious comments and advice. Those comments are all valuable and very helpful for revising and improving our paper. We have revised the manuscript accordingly, and our point-by-point responses are presented below.

Point 1: Rewrite the abstract. It should clearly state why have you taken up the work, the research design description and the results implications.

Response 1: We are grateful for the suggestion and we have rewritten the abstract, marked in red.

Point 2: Development of the constructs and hypothesis needs to the logical and adequately supported by arguments and LR.

Response 2: In order to take this concern into account, and improve the quality of our manuscript, the development of the constructs and hypothesis is supplemented in the revised version of the manuscript. We have added a subsection on Conceptual model in the Materials and Methods section of Section 2 of the paper, in which the conceptual model of the paper is supplemented. In fact, the logic of the whole study is presented throughout the whole paper. Previously, due to the page limit, we deleted some of the pages about the model and model correction from the original manuscript. In order to make the logic of the whole paper more rigorous, we put back the process of model revision and the explanation of the revision line in The Model of University-enterprise Authentic Cooperation Affecting Employment in Section 4.3, in lines 457-486.

We analyzed the path of the model in lines 358 to 405. The P value of professional theoretical knowledge to professional quality, application of professional skills and individual basic quality to professional skill application is less than 0.05, which means that these paths have reached a significant level. However, the P values of professional quality to individual basic quality and individual basic quality to application of professional skills are 0.640 and 0.193 respectively, which do not reach the significance level, so we have eliminated these two paths in the following. In the model modification stage, because the degree of freedom of the initial model is greater than 3, the correction line is added to the model according to the relevant data in the Modification Indices in the output results, and the correction line is explained in the sense of residual in lines 407-451.

Point 3: Discussions on results needs to be strengthened.

Response 3: Thank you very much for your suggestion, which is very important for our research. We have supplemented the results in lines 378-383 of the discussion, and supplemented the previous correlation in lines 397-399, which are indicated in red font.

Point 4: Limitations and implications need to be elaborated.

Response 4: We have supplemented the text according to the comments. The practical and theoretical implications of this article are adjusted and supplemented before and after the conclusion section, in lines 503-514 and lines 540-542. And we have inserted a subsection on "Limitations of the study" to the manuscript on page 14, lines 543 with red font.

Reviewer 4 Report

Dear authors,

The aspects analysed in the article are extremely interesting, i.e. to identify the factors of cooperation between universities and enterprises in an extremely interesting and actual field (cross-border e-commerce). Below I give you my recommendations for your paper: 1. There is a small discrepancy between the purpose mentioned in the abstract section and the one found in the final part of the introduction, which I would like you to clarify. 2. In the introduction section, I think that some aspects regarding the "universe" under consideration should be introduced, starting from the number of young people working in this field, the degree of absorption of graduates in this field, the number of jobs in this field, etc., in order to have a deeper overview. This will also improve the references section. 3. Line 219: please describe more precisely the concept of "representativeness". What exactly are you referring to. I assume you mean representativeness of the number of students in the university under review? Perhaps it would be better if you could clarify this statement more clearly. 4. In the discussion section, it is absolutely necessary to insert a subsection on "Limitations of the study", because I identified several during the review of the manuscript. Best regards

Author Response

Response to Reviewer 4 Comments

Dear Reviewer:

Thank you for your comment, in response to your valuable suggestions, our answers and modifications are as follows:

Point 1: There is a small discrepancy between the purpose mentioned in the abstract section and the one found in the final part of the introduction, which I would like you to clarify.

Response 1: Thank you very much for your suggestion, which is very important for our research. The purpose of this paper is to provide a more targeted value reference for the development of university-enterprise cooperation and the enhancement of students' employability by studying the factors influencing the employment of college students through university-enterprise true cooperation, so as to further promote the in-depth implementation of the integration of industry and education. And we have rewritten the abstract and supplemented the conclusion section with red font.

Point 2: In the introduction section, I think that some aspects regarding the "universe" under consideration should be introduced, starting from the number of young people working in this field, the degree of absorption of graduates in this field, the number of jobs in this field, etc., in order to have a deeper overview. This will also improve the references section.

Response 2: We have included an overview of the industry background, graduate employment, and university situation in lines 41-54 of the introduction, marked in red.

Point 3: Line 219: please describe more precisely the concept of "representativeness". What exactly are you referring to? I assume you mean representativeness of the number of students in the university under review? Perhaps it would be better if you could clarify this statement more clearly.

Response 3: Thanks for your comments, the discussion regarding this question is presented following:

Since the targets of this survey are college students of cross-border e-commerce majors and enterprises of cross-border e-commerce industry in Hangzhou undergraduate colleges and universities, we designed the research questionnaires for students and enterprises respectively to conduct the survey. Hangzhou is the first city in China to carry out a comprehensive experimental zone for cross-border e-commerce, and Zhejiang International Studies University is a model institution for cross-border e-commerce talent training in China. The research subjects we selected were students and enterprises participating in the double-selection meeting of Zhejiang International Studies University, which is one of the earliest universities in China to develop cross-border e-commerce majors, and the companies docked with them in the double-selection meeting also include a large number of cross-border e-commerce enterprises in Zhejiang. Therefore, the "representativeness" mentioned here means that the students and enterprises are representative of the cross-border e-commerce profession and industry in Zhejiang, and since both are from the double-selection, the two types of data are also comparable. We clarify "representativeness" more clearly in lines 262-273 with red font.

Point 4: In the discussion section, it is absolutely necessary to insert a subsection on "Limitations of the study", because I identified several during the review of the manuscript.

Response 4: Thank you for your suggestion. We have inserted a subsection on "Limitations of the study" to the manuscript on page 14, lines 543 with red font.

Round 2

Reviewer 1 Report

The second round comment is in green. Please respond to them 

Response to Reviewer 1 Comments

Dear Reviewer:

We are very grateful to your comments for the manuscript. According with your advice, we amended the relevant part in manuscript. Some of your questions were answered below.

Point 1: The gap should be based on previous papers. It would be useful to make this more obvious. Clearly, state which paper made calls for such a study.

Response 1: We are grateful for the suggestion. Our research is inspired by the current situation of cross-border e-commerce university-enterprise cooperation, and an article in the journal of Education and Training titled Can university-industry linkages stimulate student employability also provides ideas for our research. The content in red font on lines 151-153 is a more detailed supplement.

You have mentioned Ishengoma and Vaaland, but are they making calls for further studies !! If yes, state that clearly. And are the only study calling for that? And you have cited the source mistakenly without putting the year for it?

Point 2: Do you have ethical approval for this research? If yes, state it in this methodology section.

Response 2: Thank you very much for your suggestion, which is very important for our research. In fact, we have made a statement at the beginning of each survey questionnaire: we will keep the information strictly confidential and promise to use it only for scientific research and policy recommendations, not for other purposes. All respondents answered the questionnaire after accepting the commitment. We have made supplementary explanations in lines 282-285 of the text, marked in red.

Accepted

Point 3: I was wondering about your sample size, 208! So, does your sample represent all the students or enterprises worldwide? If it is all the students worldwide, how can you ensure that your sample size represents all students worldwide!! Or is it specific to a certain geographical area or country? Please state which geographical or country in which this study was conducted!! As students’ characteristics might differ from country to country, and state how you have achieved validity and reliability, you might also need to reflect that in the title.

Response 3: Thanks for your comments, the discussion regarding this question is presented following: The subjects of this survey are undergraduates majoring in cross-border e-commerce in universities in Hangzhou and enterprises in the cross-border e-commerce industry, which is mentioned on lines 273-278.

Since the targets of this survey are college students of cross-border e-commerce majors and enterprises of cross-border e-commerce industry in Hangzhou undergraduate colleges and universities, we designed the research questionnaires for students and enterprises respectively to conduct the survey. Hangzhou is the first city in China to carry out a comprehensive experimental zone for cross-border e-commerce, and Zhejiang International Studies University is a model institution for cross-border e-commerce talent training in China. The research subjects we selected were students and enterprises participating in the double-selection meeting of Zhejiang International Studies University, which is one of the earliest universities in China to develop cross-border e-commerce majors, and the companies docked with them in the double-selection meeting also include a large number of cross-border e-commerce enterprises in Zhejiang. Therefore, the "representativeness" mentioned here means that the students and enterprises are representative of the cross-border e-commerce profession and industry in Zhejiang, and since both are from the double-selection, the two types of data are also comparable. We clarify "representativeness" more clearly in lines 262-273 with red font.

I would suggest that you change the title of your research

An Empirical Study of the Influencing Factors of University- 2 enterprise Authentic Cooperation on Cross-border E-commerce 3 Employment: The case of Zhejiang

Point 4: What are the practical and theoretical implications for your paper? Please state these before and after the conclusion section.

Response 4: We have supplemented the text according to the comments. The practical and theoretical implications of this article are adjusted and supplemented before and after the conclusion section, in lines 503-514 and lines 540-542, marked in red.

Accepted. 

Author Response

Response to Reviewer 1 Comments

Dear Reviewer:

Thank you for your suggestion, and our further reply is as follows:

Point 1: The gap should be based on previous papers. It would be useful to make this more obvious. Clearly, state which paper made calls for such a study.

Response 1: We are grateful for the suggestion. Our research is inspired by the current situation of cross-border e-commerce university-enterprise cooperation, and an article in the journal of Education and Training titled Can university-industry linkages stimulate student employability also provides ideas for our research. The content in red font on lines 151-153 is a more detailed supplement.

You have mentioned Ishengoma and Vaaland, but are they making calls for further studies !! If yes, state that clearly. And are the only study calling for that? And you have cited the source mistakenly without putting the year for it?

Response: Can university industry links stimulate student employment mainly provides inspiration and research ideas for our research. Ishengoma and Vaaland pointed out at the end of the paper that not all developing countries have rich natural resources hold a strategic position sufficient to motivate MNEs to invest in UILs. And they believe that this weakness is considered as a minor point in the overall set of findings. Other papers cited by us have also emphasized the importance of continuing to study the mechanism of university-enterprise cooperation affecting students' employability. It provides us with a lot of inspiration on research ideas and methods.

The year of the paper is indicated in the references: Ishengoma, E.; & Vaaland, T. I. Can university-industry linkages stimulate student employability?. Education+ training 2016, 58, 18-44.

Point 3: I was wondering about your sample size, 208! So, does your sample represent all the students or enterprises worldwide? If it is all the students worldwide, how can you ensure that your sample size represents all students worldwide!! Or is it specific to a certain geographical area or country? Please state which geographical or country in which this study was conducted!! As students’ characteristics might differ from country to country, and state how you have achieved validity and reliability, you might also need to reflect that in the title.

Response 3: Thanks for your comments, the discussion regarding this question is presented following: The subjects of this survey are undergraduates majoring in cross-border e-commerce in universities in Hangzhou and enterprises in the cross-border e-commerce industry, which is mentioned on lines 273-278.

Since the targets of this survey are college students of cross-border e-commerce majors and enterprises of cross-border e-commerce industry in Hangzhou undergraduate colleges and universities, we designed the research questionnaires for students and enterprises respectively to conduct the survey. Hangzhou is the first city in China to carry out a comprehensive experimental zone for cross-border e-commerce, and Zhejiang International Studies University is a model institution for cross-border e-commerce talent training in China. The research subjects we selected were students and enterprises participating in the double-selection meeting of Zhejiang International Studies University, which is one of the earliest universities in China to develop cross-border e-commerce majors, and the companies docked with them in the double-selection meeting also include a large number of cross-border e-commerce enterprises in Zhejiang. Therefore, the "representativeness" mentioned here means that the students and enterprises are representative of the cross-border e-commerce profession and industry in Zhejiang, and since both are from the double-selection, the two types of data are also comparable. We clarify "representativeness" more clearly in lines 262-273 with red font.

I would suggest that you change the title of your research

An Empirical Study of the Influencing Factors of University- 2 enterprise Authentic Cooperation on Cross-border E-commerce 3 Employment: The case of Zhejiang

Response: Thank you very much for your suggestion and we have revised the title of the paper.

Reviewer 3 Report

I dont have any further comments

Author Response

The second round of review did not receive any comments that need to be modified. Thank you for your comments and affirmation of our work!

Round 3

Reviewer 1 Report

Comment in Blue 

Response 1: We are grateful for the suggestion. Our research is inspired by the current situation of cross-border e-commerce university-enterprise cooperation, and an article in the journal of Education and Training titled Can university-industry linkages stimulate student employability also provides ideas for our research. The content in red font on lines 151-153 is a more detailed supplement.

You have mentioned Ishengoma and Vaaland, but are they making calls for further studies !! If yes, state that clearly. And are the only study calling for that? And you have cited the source mistakenly without putting the year for it?

This takes us back to the main point I suggested in the first round of the review. I think the gap in your study should be based on a number of studies, not only one paper. Furthermore, you should clearly state the shortcoming of these studies and what sort of calls these papers make to justify your study. 

Author Response

Response to Reviewer 1 Comments

Dear Reviewer:

Thank you for your suggestion, and our further reply is as follows:

Point 1: This takes us back to the main point I suggested in the first round of the review. I think the gap in your study should be based on a number of studies, not only one paper. Furthermore, you should clearly state the shortcoming of these studies and what sort of calls these papers make to justify your study. 

Response 1: The research gap that we have found is indeed based on a number of studies, and for clarity, we have listed a table. The table includes the main studies we have cited for this study, as well as the views of these studies, their shortcomings, and the inspiration they have given us. The last page of the table shows where these studies appear in our paper.

File Name

Key work

The gap we based, the shortcoming we found and the inspiration we got

Page numbers in the paper

University–industry linkages in the UK: What are the factors underlying the variety of interactions with industry?

Este and Patel

(2007)

This paper examines the different channels of interaction between academic researchers and industry through a large-scale survey of British academic researchers, as well as the factors that affect their participation in various interactions. The research complements the model of interaction between university researchers and industry, which has provided us with inspiration.

Page 2

Lines 69-73

Towards a new model of EU-China innovation cooperation:

Bridging missing links between international university collaboration and international industry collaboration

Yuzhuo Cai

(2023)

The research in this paper mainly focuses on the transnational level. They find that international university collaboration could contribute to international industry collaboration. It studies how to promote university-enterprise cooperation to maximize revenue and minimize risk. The conclusion of the study is based on the EU-China context, and other contexts need to be discussed separately. Therefore, we conducted a microscopic study at the urban level.

Page 2

Lines 80-84

Effects of Local Government Behavior on University–Enterprise Knowledge Flow:

Evidence from China

Shaopeng Zhang

and

Xiaohong Wang

(2022)

This paper divides the knowledge flow between universities and enterprises into two stages: knowledge creation and knowledge transfer, further promoting the research progress of university-enterprise cooperation in the field of knowledge management. The author points out that the sample of this study involves 30 provinces in China, and the research scope is relatively macro, so there is a lack of pertinence. Future research can narrow the scope of research to the urban level. Inspired by this, this article selects major universities in Hangzhou, Zhejiang Province, for research, focusing on the micro level.

Page 2

Lines 86-92

Investigating the factors that diminish the barriers to university-industry collaboration

Bruneela et al.

(2010)

This paper explores how to reduce barriers in university-enterprise cooperation. Based on large-scale surveys and public records, it explores the impact of cooperation experience, breadth, and trust between organizations on reducing barriers. Research has found that trust between organizations is one of the strongest mechanisms for reducing barriers to university-enterprise cooperation and interaction. This paper conducts research from the perspective of barriers to university-enterprise cooperation, which makes us wonder what are the barriers to student employment? How can university-enterprise cooperation eliminate these obstacles?

Page 2

Lines 93-95

Graduate Employability: A Review of Conceptual and Empirical Themes

Michael Tomlinson

(2012)

The paper explores some of the conceptual notions that have informed understandings of graduate employability, and provides inspiration for the definition of relevant concepts in our research. Through this series of definitions, it can be seen that employability not only refers to the result of seeking a job, but also emphasizes the potential of individuals to develop and succeed in the industry field.

Page 3

Lines 122-124

A suggested best practices for enhancing performance of soft skills with entry-level hospitality managers

Weber et al.

(2020)

This study explores the key soft skills that affect employees' further development through how to add soft skills to company performance evaluations to help company managers and employees improve their performance. The importance of specific soft skills training for managers was emphasized. It focuses on the importance of improving soft skills training for managers and employees for company performance, mainly providing us with inspiration on relevant soft skills. Students also need these soft skills in employment, and university-enterprise cooperation is a good opportunity to cultivate these soft skills. The definition of soft power given in this article also provides inspiration for us to construct relevant indicators.

Page 3

Lines 126-128

Business’ students industrial training: Performance and employment opportunity

Erni bte Tanius

(2015)

This paper reports the perception of 187 industrial supervisors on the 307 business students’ performance and their employment opportunity during industrial training. The results show that the students’ performances are excellent in the areas punctuality, honesty, teamwork, and relations with colleagues. Many previous studies believed that internships can bring benefits to both students and enterprises. From the perspective of employability, internships can improve students' employability skills, thus improving employability and gaining favor from enterprises. Moreover, in the process of internship, students can obtain real employment experience, and enterprises can also find potential employees.

Page 3

Lines 133-137

Can university-industry linkages stimulate student employability?

Ishengoma

and  

Vaaland

(2016)

The study found that university-enterprise cooperation in the form of internships for students in enterprises and lectures by enterprises in universities can significantly improve the employability of students. Similar research is mostly conducted in the context of developed countries, and this paper complements such research by conducting research in the context of developing countries. Ishengoma and Vaaland pointed out at the end of the paper that not all developing countries have rich natural resources hold a strategic position sufficient to motivate MNEs to invest in UILs. It mainly provides inspiration and research ideas for our research.

Page 4

Lines 157-159
